# Integral Sliding Control Approach for Generalized Cyclic Pursuit Formation Maintenance

**Antoine Ansart** *,† and **Jyh-Ching Juang** †

Department of Electrical Engineering, National Cheng Kung University, Tainan 701, Taiwan; juang@mail.ncku.edu.tw
* Correspondence: ansartantoine0@gmail.com or antoineansart@yahoo.fr; Tel.: +886-909-210-493
† These authors contributed equally to this work.

**Abstract:** This paper is concerned with the formation maintenance of a group of autonomous agents under generalized cyclic pursuit (GCP) law. The described pattern for agents under such formation is epicycle-like. For a network of agents to achieve such a formation, marginal stability of the overall network is required. The desired marginal stability of the network relies on each agents' gain values, and uncertainties in these gains can occur. Previous studies have used fixed gains, we enhance the stability of the gains via a dynamic approach using an integral sliding controller (ISC). An ISC can ensure sliding behavior of the gains throughout the entire response, and it is shown that the gains are robust toward variations and thus make the network keep its marginal stability and its formation.

**Keywords:** control theory; cyclic pursuit; dynamic gain; formation control; marginal system; multi-agent systems; sliding control





## 1. Introduction

Multi-agent systems (MAS) are composed of a group of agents interacting under a cooperation protocol. The essential focus of MAS literature is made on obstacle avoidance as well as formation control and maintenance. Formation control is not new as it presents important practical and theoretical significance, and we direct the reader to [1] for a recent survey regarding multi-agent formation control. Among the formation control approaches, cyclic pursuit is one of the most studied [2–8]. A cyclic pursuit scheme is defined by the representation of a group of $n$ leaderless agents, labeled by a number, and where agent $i$ pursues its leading neighbor labeled $i + 1$ (the pursuit), modulo $n$ (the cycle). A part of the cyclic pursuit research concentrates on formation around a fixed or moving target.

On one side, researchers focus on agents that maintain a constant or dynamic inter-agents spacing while rotating around a target. This circular formation pattern is of interest when surveillance, or observation, is defined as the objective [6,9–14]. One of the specific properties of circular formation is that constant values can be used by agents in order to correct their gains and hence maintain their formation [12]. As at least one value is fixed (e.g., inter-agent distance, or distance to a target), it can be used to compute the error and thus correct the gains of the agents.

On the other hand, researchers have focused on non-circular pattern generation. By employing a single agent, a plethora of patterns were achieved by Sinha A., Tripathy T. et al. throughout various approaches and algorithms [15–19]. By deploying more agents under a general cyclic scheme, it was possible to generate epicycle or trochoidal-like patterns for a groups of agents modeled as single integrator [20–23]; which is of interest in the present paper. In these papers, the control law developed to reach epicycle or trochoidal patterns uses fixed-gain methods. The analysis of the overall stability of the system leads the authors to the fact that to reach such motion, the overall system has to be marginally stable, which is the key problem that we propose to solve in this paper. As will be seen in Section 2, the eigenvalue repartition relies on the gains of the agents, and with fixed

value, no correction of the gains is possible. A major difference with circular formation is the fact that, under a more generalized cyclic pursuit (GCP) scheme, as the inter-agent distance and the distance to the target are time-varying, no method using a constant value as a reference to correct the positions or the gains can be used.

To address this problem and thus sustain the formation, the method here focuses on a dynamic gains approach. To name only a few results, the use of adaptive methods for cyclic pursuit can already be found in [24], where the authors use time-varying feedback coupling gains in order to synchronize the evolution of a complex network. In addition, the work conducted in [25] deals with synchronization of MAS by using weightage based on local information. Recently, the authors in [26,27] use model reference adaptive control on a MAS under generalized cyclic pursuit scheme to maintain the formation despite the present uncertainties.

Dynamic gains approach to maintain circular formation combined with the idea of generating epicycle-like pattern constitute the backbone of our contribution. Agents are homogeneous, meaning they all have the same control gain, selected properly to lead the agents to behave in an epicycle way. We bring our attention to the gains themselves and turn them into dynamic versions. In this dynamic version, the gains correct themselves and are robust to uncertainties. Thus, the problem can be reduced to a one-dimensional problem. As the formation patterns depend on the initial positions of agents and their gains' value, it is mandatory to fix the gains' value to their final value at the fastest pace possible. Sliding control or sliding mode control (SC) is a well-known topic, its robustness reputation is now well-established, and by adding an integral term, the SC is turned into integral sliding mode controller (ISC). The curious reader is refer to [28–32] for reviews of sliding control under a mathematical approach and its modified types for robotic application as well as literature about the ISC. To our knowledge, no such approach has been yet developed.

We first state the problem with a mathematical explanation in a more developed form. The proposed gains control law will then be derived and detailed. Simulations showing the pertinence of this law will then be exposed, and the conclusion and remarks will close the paper.

**Notation 1.** *The bold mathematical letters define matrix and/or vector. $m$ is the dimension of the network ($m = 2$ for plane and $m = 3$ for space representation). The subscript $_m$, $_n$ and $_{2n}$ are added to help the reader with the size of the squared matrices. The subscript $_f$ indicates the final value of the variable of interest. The superscript $^T$ means transpose. The bold letter $\boldsymbol{i}$ stands for the imaginary number such that $\boldsymbol{i}^2 = -1$; it should not be mistaken with i which represent the agents' identification number. The words pattern/formation are interchangeable as they are used in the same sense.*

## 2. Problem Statement

Consider a fleet of $n$ agents in the plane, where every agent is aware of its own position relative to a fixed target with fixed axis (absolute position) and the position of its leader (relative position, or measurement).

Suppose a target located at the origin of an absolute frame of reference. The aim is to let the fleet of agents to surround this target and simultaneously cover an annulus defined by an inner and outer radii, $r_{\text{in}}$ and $r_{\text{out}}$, respectively. The design constraints being defined, it will bring some design parameters that have to be respected, namely $a$ and $b$ that will be developed further. By using an appropriate design model for each agents, the specific gains, based on the constraints, will be deduced. A graphical representation of the design constraints and the parameters is depicted in Figure 1. Moreover, those agents will be under a general cyclic pursuit law for their motion to be epicycle.

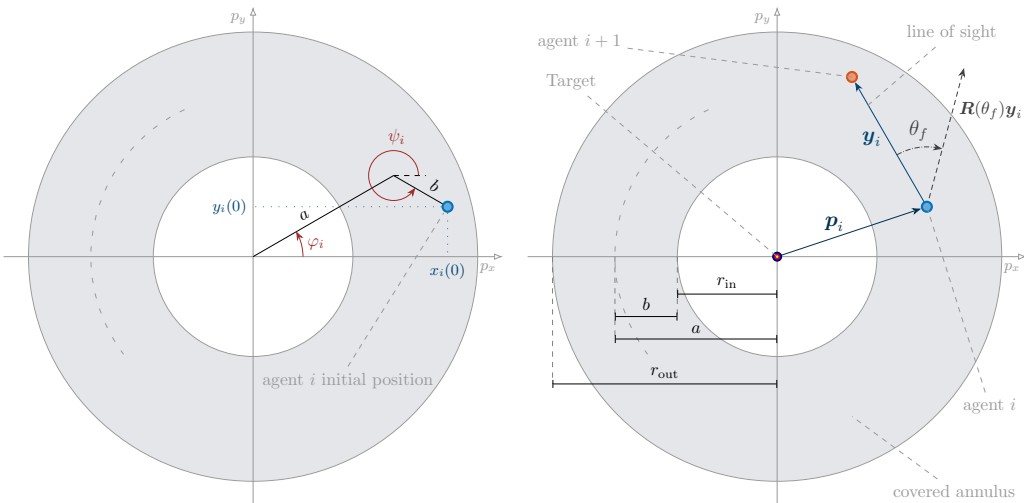

**Figure 1.** Agents in the 2-D plane and annulus parameters.

First, the kinematics of each agents is described as

$$\dot{\boldsymbol{p}}_i = \boldsymbol{r}_{p,i}\,, \quad i = 1, 2, ..., n \tag{1}$$

where $\boldsymbol{p}_i = \begin{bmatrix} p_{x,i} & p_{y,i} \end{bmatrix}^{\mathrm{T}}$ is the absolute position vector of agent $i$ in the 2-D plane and $\boldsymbol{r}_{p,i}$ is its control law defined as

$$\boldsymbol{r}_{p,i} = \boldsymbol{R}(\theta_f)\boldsymbol{y}_i - g_p \boldsymbol{p}_i \tag{2}$$

where

$$\boldsymbol{y}_i = \begin{bmatrix} p_{x,i+1} - p_{x,i} \\ p_{y,i+1} - p_{y,i} \end{bmatrix} \tag{3}$$

is the measurement vector of agent $i$. Note that every agent obeys the same law (2) and, by consequence, the constant gains defined by $\boldsymbol{R}(\theta_f)$ and $g_p$ have to be homogeneous. Define the line of sight (LoS) as the line from an agent's "eye" to its leader and $\theta_f$ as the common clockwise rotation angle of this LoS, then its rotation matrix is

$$\boldsymbol{R}(\theta_f) = \begin{bmatrix} \cos\theta_f & \sin\theta_f \\ -\sin\theta_f & \cos\theta_f \end{bmatrix} \tag{4}$$

The following selection of the gains

$$\theta_f = (2h^* + 1)\frac{\pi}{n}\,, \quad h^* = 0, 1, \cdots, n-1 \tag{5a}$$

$$g_p = \cos\left(\frac{\pi}{n}\right) - \cos(\theta_f) \tag{5b}$$

will bring the agents to describe an epicycle-like motion. It can be seen from (5) that $\theta_f$ is the key parameter that gives the value of the gains. The parameter $h^*$ has also to be chosen according to the type of desired motion. Table 1 (taken from [20]) summarizes the link between the couple $n$, $h^*$ and the possible formations. However, circle and ellipse cases by choice of $h^*$ are degenerate cases and will not be considered. It has to be noted that the case $h^* = 1$ is always possible as long as $n > 4$ and, hence, it is assumed in the rest of the paper that $h^* = 1$.

**Table 1.** Relationship between the patterns and $(n, h^*)$.

| Formation | $(n, h^*)$ |
|---|---|
| Circle | $\forall n,\ h^* = 0$ or $n-1$ |
| | $n$ even, $h^* = \frac{n}{2} - 1$ or $\frac{n}{2}$ |
| Ellipse | $n$ odd, $h^* = \frac{n-1}{2}$ |
| Algebraic | $(6, 1), (6, 4)$ |
| Transcendental | $(5, 1), (5, 3), (7, 1), (7, 2), (7, 4), (7, 5), (8, 1), (8, 2), (8, 5), (8, 6), \ldots$ |

The overall dynamic of the network is then

$$\dot{x} = Ax \tag{6}$$

where $x = \begin{bmatrix} p_1^T & p_2^T & \cdots & p_n^T \end{bmatrix}^T$ is the stacked position vector, and

$$A = -g_p I_{2n} - W_n \otimes R(\theta_f) \tag{7}$$

is the state matrix that will determine the stability of the system.

$W_n$ is the following circulant matrix

$$W_n = \begin{bmatrix} 1 & -1 & \ldots & 0 \\ 0 & 1 & \ldots & 0 \\ \vdots & \vdots & \ddots & \vdots \\ -1 & 0 & \ldots & 1 \end{bmatrix} \tag{8}$$

and "$\otimes$" represents the Kronecker product. Definitions and properties for circulant matrices and Kronecker product are given in Appendix A.

The properties of the Kronecker product coupled with the ones of circulant matrix show that the eigenvalues of (7) are

$$\lambda_j = -g_p - \left(1 - \exp\left(i\frac{2\pi j}{n}\right)\right) \exp\left(\pm i\theta_f\right), \quad j = 1, \cdots, n \tag{9}$$

Putting (5) into (9), the computations reveals that two specifications has to be noted

1. The overall network posses two pairs of imaginary-axis eigenvalue expressed as

$$\lambda_a = \pm i\omega_a, \quad \lambda_b = \pm i\omega_b \tag{10}$$

where

$$\omega_a = \sin\theta_f - \sin\frac{\pi}{n}, \quad \omega_b = \sin\theta_f + \sin\frac{\pi}{n} \tag{11}$$

and thus epicycle motion is rendered.

2. By developing (9), it can be seen that the real and imaginary parts of the eigenvalues rely on $\theta_f$.

Based on the ideas presented in [20,33], we enhance the robustness of the system by extending the static-gain control law to a dynamic-gain control law. By using (2), all agents experienced an epicycle motion, but it has to be noted that agents can also render trochoid motion. This extension has been made in [22], where trochoidal pattern formation are detailed; examples of epicycle and trochoidal-like formation are depicted in Figure 2.

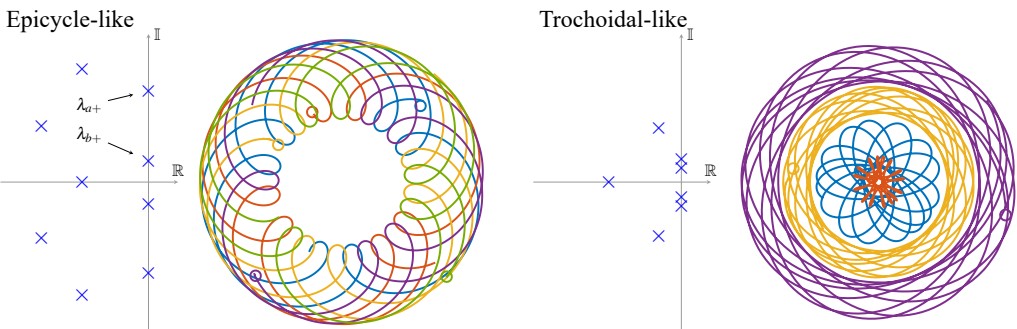

**Figure 2.** Examples of eigenvalue repartition and the corresponding pattern of agents.

The formation pattern relies on the eigenvalue repartition, and the eigenvalues themselves rely on the common value of $\theta_f$ in (5a); hence, it is the key determinant for the motion to be sustained. As with a fixed value, uncertainties can occur, and the overall network is prone to being unstable. To sustain the formation, the following dynamic method is proposed.

### 3. Dynamic Gain Proposal

As stated, $\theta_f$ is a value that can be subjected to uncertainties. In addition, this value must be the same for all agents. This value is leading the gain so it is considered in the following that the terms *gains* and *angle* are interchangeable. The gains thus have to be resistant to uncertainties. We define $\theta_f$ as the value that all agent angles must tend to. Variations in the angle can happen at any time under any form; however, only matching uncertainties are taken into account by assumption, and the following theorem is stated.

**Theorem 1.** *If a network of n agents modeled by a single integrator are under the control law (2) in which the gains of each agent are set to*

$$\ddot{\theta}_i = u_i + d_i \tag{12a}$$

$$g_{p,i} = \cos\left(\frac{\pi}{n}\right) - \cos(\theta_i) \tag{12b}$$

*with $d_i$ a bounded disturbance, and $u_i$ the command defined as*

$$u_i = -\kappa_p\left(\theta_i - \theta_f\right) - \kappa_v\dot{\theta}_i - m_i sign(s) \tag{13a}$$

$$s = \sigma + z \tag{13b}$$

*where $\kappa_p$, $\kappa_v$, $m_i$ are positive value, s is the sliding surface in which $\sigma$ may be designed as a linear combination of the system dynamic angles and z is designed to be seen as an integrator, then at steady-state the eigenvalues of (7) will possess two pairs of imaginary axis eigenvalues and the network (6) will keep its marginal stability despite uncertainties in agents' gains.*

**Proof of Theorem 1.** First of all, considering $d_i = 0$ and steady-state, $\theta_i$ in (12a) tends to $\theta_f$ in (5a) and thus rejoins the statement linked with Equation (10); thus the eigenvalues' repartition is ensured.

The choice of (13) will be demonstrated here, where $u_i$ will be developed into two parts

$$u_i = u_{0,i} + u_{d,i} \tag{14}$$

with $u_{0,i}$ being the nominal control and $u_{d,i}$ being designed to reject the disturbance term.

Start by establishing (12a) in state-space form

$$\dot{\boldsymbol{\omega}}_i = \boldsymbol{\Lambda}\boldsymbol{\omega}_i + \boldsymbol{\Pi}u_i + d_i(\boldsymbol{\omega}_i, t) \tag{15}$$

with $u_i$ the control gain law, the following matrices

$$\boldsymbol{\omega}_i = \begin{bmatrix} \theta_i \\ \dot{\theta}_i \end{bmatrix}, \quad \boldsymbol{\Lambda} = \begin{bmatrix} 0 & 1 \\ 0 & 0 \end{bmatrix}, \quad \boldsymbol{\Pi} = \begin{bmatrix} 0 \\ 1 \end{bmatrix}, \tag{16}$$

and $d_i$ the bounded disturbance function of $\boldsymbol{\omega}_i$ and $t$. The disturbance comprises the perturbation owed to parameter variations, unmodeled dynamics, and external disturbances and is assumed to fulfill the matching condition; that is, perturbations that enter the state equation at the same point as the control input, i.e.,

$$d_i = \boldsymbol{\Pi} u_{\delta,i}$$

with

$$|d_i| \leq L_i$$

where $L_i$ being known positive value.

At first, consider that no disturbances occur in (15). By so, the nominal control is chosen as

$$u_{0,i} = -\kappa_p \left( \theta_i - \theta_f \right) - \kappa_v \dot{\theta}_i \tag{17}$$

in which $\kappa_p$ and $\kappa_v$ are fixed gains appropriately chosen for the heading angle to reach $\theta_f$ in desired time. Equation (17) would give the perfect heading angle dynamic

$$\dot{\boldsymbol{\omega}}_{0,i} = \boldsymbol{\Lambda}\boldsymbol{\omega}_{0,i} + \boldsymbol{\Pi} u_{0,i} \tag{18}$$

$u_{0,i}$ is not robust enough to abrogate the effects of uncertainties. Hence, the addition of $u_{d,i}$. To satisfy $\boldsymbol{\omega}_i \equiv \boldsymbol{\omega}_{0,i}$, the perfect control of $u_{d,i}$, denoted $u_{p,i}$, has to fulfill

$$\boldsymbol{\Pi} u_{p,i} = -d_i$$

in other words

$$u_{p,i} = -u_{\delta,i} \tag{19}$$

and hence $u_{p,i}$ will perfectly describe the angles' trajectories when sliding along $s(\boldsymbol{\omega}_i) = 0$ in (13b). Replacing (14) in (15) provides

$$\dot{\boldsymbol{\omega}}_i = \boldsymbol{\Lambda}\boldsymbol{\omega}_i + \boldsymbol{\Pi}(u_{0,i} + u_{d,i}) + d_i(\boldsymbol{\omega}_i, t)$$

and by recalling the sliding surface (13b), the sliding manifold $\sigma$ and the integral term has to be developed. The following manifold $\sigma$ is selected

$$\dot{\sigma}(\boldsymbol{\omega}_i) = \frac{\partial \sigma}{\partial \boldsymbol{\omega}_i} \dot{\boldsymbol{\omega}}_i$$

in which we assume that $\frac{\partial \sigma}{\partial \boldsymbol{\omega}_i} \boldsymbol{\Pi}$ is non-singular.

The achievement of (19) is based on the choice of $z$ and the condition $\dot{s}(\boldsymbol{\omega}_i) = 0$ has to be reached; thus from (13b)

$$0 = \dot{\sigma}(\boldsymbol{\omega}_i) + \dot{z} \tag{20}$$

choosing

$$\dot{z} = -\frac{\partial \sigma}{\partial \boldsymbol{\omega}_i}(\boldsymbol{\Lambda}\boldsymbol{\omega}_{0,i} + \boldsymbol{\Pi} u_{0,i}) \tag{21}$$

and replacing (21) in (20), one gets (19). It has to be noted that $z(0)$ is based on the fact that $s(0) = 0$, i.e., where sliding mode occurs from the initial time moment. As (19) is satisfied, the motion of the angles in sliding mode will be like (18), in which the disturbances do not exist.

Stating

$$u_{d,i} = -m_i \text{sign}(s) \tag{22}$$

where $m_i$ is a constant specifying the converging rate of $s$ selected in accordance to

$$m_i > L_i,$$

the sliding mode can be enforced in the surface $s = 0$, where the disturbances are rejected. Putting (22), (21) into the derivative of $s$ gives

$$\dot{s} = \frac{\partial \sigma}{\partial \boldsymbol{\omega}_i} \boldsymbol{\Pi} \left( u_{\delta_i} - m_i \text{sign}(s) \right)$$

and hence gives the law (13).

By using the aforementioned (13), the sliding mode of the gains is enforced on $s = 0$ and uncertainties in the heading angle gain are rejected for every agent.

As the network's formation relies on the eigenvalues' repartition, and the eigenvalues themselves rely on the angle's gains, the overall system keeps its marginal stability.　□

An important part of the heading angle law is the discontinuity. The constant switching will lead to high frequency oscillations. By replacing (14) with

$$u_i = u_{0,i} + u_{q,i} \tag{23}$$

where

$$\dot{u}_{q,i} = -\frac{m_i}{\epsilon} \text{sign}(s) - \frac{1}{\epsilon} u_{q,i} \tag{24}$$

with $\epsilon$ is a small time constant [34], the chattering can be eliminated as we use a low-pass filter on (22) in the proof of theorem 1. Yet, decreasing $\epsilon$ may lead to chattering in the control law. We will now demonstrate our method on some specific simulations.

## 4. Simulations

A group of $n = 5$ agents has the aim of describing an epicycle motion while enclosing a target located at the origin. The covered area is defined according to the inner and an outer radii, and as explained in [20,33], the initial positions of agents are related to the constraints parameters by

$$\boldsymbol{p}_{i,0} = a \begin{bmatrix} \cos(\varphi_i) \\ \sin(\varphi_i) \end{bmatrix} + b \begin{bmatrix} \cos(\psi_i) \\ \sin(\psi_i) \end{bmatrix}$$

where

$$r_{\text{in}} = a - b \qquad\qquad \varphi_i = 2h^*(i-1)\frac{\pi}{n}$$

$$r_{\text{out}} = a + b \qquad\qquad \psi_i = 2(h^*+1)(i-1)\frac{\pi}{n}$$

and $\varphi_i$ and $\psi_i$ are covered area design parameters; see Figure 1.

The agents are thus subject to the motion described by (6) and (7), in which $g_{p,i}$ is expressed in (12b). Each agents' dynamic angle follows (12a) where $u_i$ is expressed in (13) and $d_i$ is selected to be

$$d_i(t) = \delta_i(\sin(\omega_i t) + U(t_i))$$

where $\delta_i$, $\omega_i$ are constants appropriate to each agent, and $U(t_i)$ is a step disturbance happening at time $t_i$. The following values are selected for the simulation

$$r_{\text{out}} = 20, \quad r_{\text{in}} = 9, \quad h^* = 1$$

and the results are shown in Figures 3 and 4.

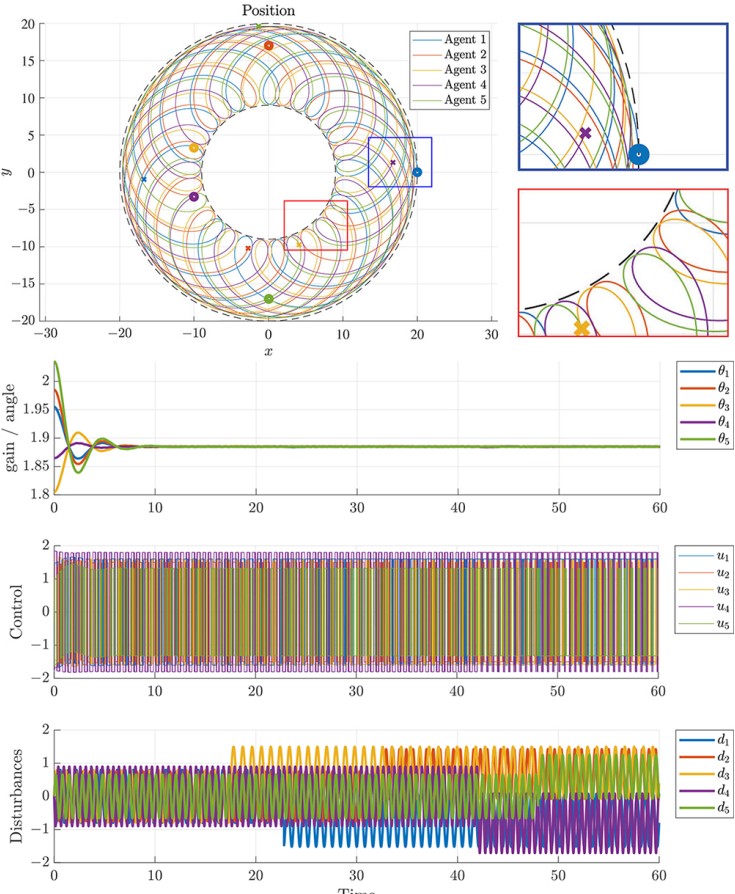

**Figure 3.** Pattern and angle variations, control, and disturbances under (13).

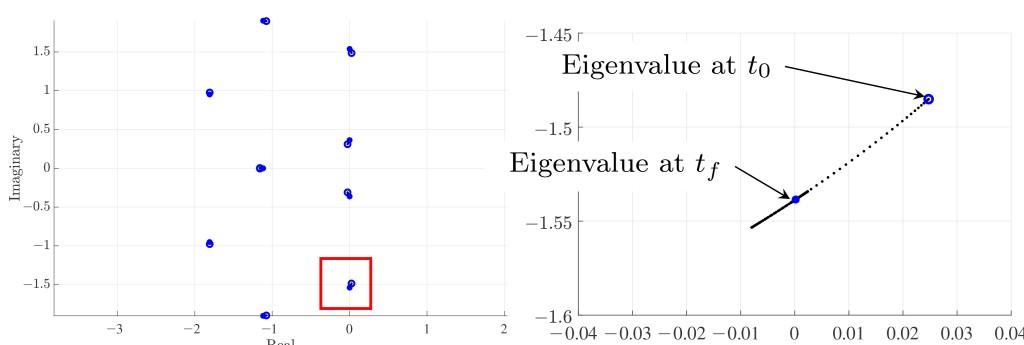

**Figure 4.** Eigenvalue repartition for the first simulation.

At $t = 0$, all agents angle are set to be $\theta_i(0) = \theta_f \pm 0.15$. Thus, the overall network eigenvalues are not yet properly set; the eigenvalues of interest present a positive real part as shown in Figure 4. As theta is dynamic, its response is shown in Figure 3, and the theta of all agents tends to the same common value computed by (5a); in this example

$$\theta_f = \frac{3\pi}{n} \approx 1.885$$

uncertainties present in the gains are also shown in Figure 3 along with the control effort. It can be seen that the uncertainties are perfectly rejected. In addition, it has to be noted that agents stay in the specified annulus, with an acceptable margin as shown in Figure 3. The dashed lines represent the bounds of the covered area and it is seen that agents stay within these bounds. The settling time needed for heading angles to settle down have an

impact on those margins. If every angle initial value were set perfectly and agents' initial positions also perfect, the area would be covered perfectly within the bounds.

As a large control effort is required to counterattack the disturbances, a second simulation using (23) and (24) is depicted in Figure 5.

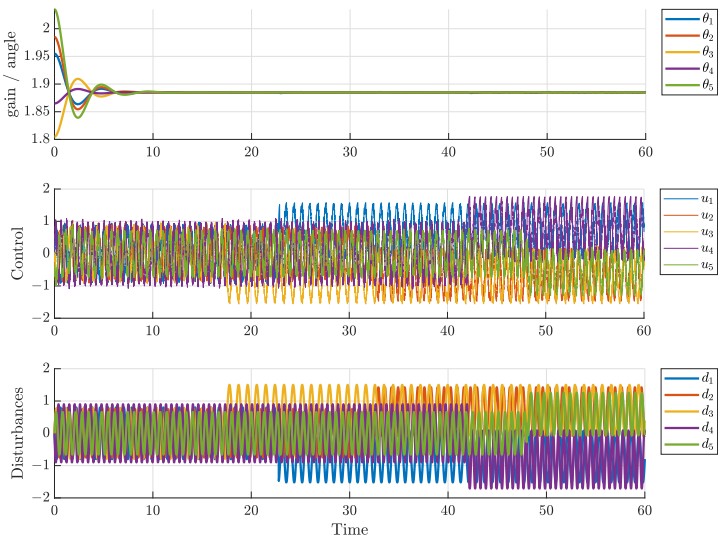

**Figure 5.** Angle variations, control, and disturbances under (23).

It can be seen that the control effort is largely reduced compared to the previous simulation while keeping the angle to the desired value.

## 5. Conclusions and Future Work

In this paper, a group of homogeneous agents is deployed to surround a target and cover a specific annulus around it. Using a generalized cyclic pursuit scheme, agents describe epicycle-like motion to cover the area. The eigenvalue repartition of the overall network reveals the presence of purely imaginary axis eigenvalues that are necessary to ensure the formation to be described. As the eigenvalue repartition relies on the gains of the agents, we propose to ensure the robustness of these gains via the use of an integral sliding control. Uncertainties may arise in the gains, and the ISC rejects those uncertainties and ensures, then, that the network keeps its formation.

Further improvements are still to be made, such as in formation pattern diversity or a more optimal choice in $u_{0,i}$. Moreover, enhancement in the choice of the ISC could lead to even more robust controllers.

**Author Contributions:** A.A. conceptualized and performed the experiments, analyzed the data and wrote the manuscript. J.-C.J. supervised the research, providing guidance theory, analyzed the data, and revised the paper. Both authors have read and agreed to the published version of the manuscript.

**Funding:** The research is supported by the Ministry of Science and Technology (MOST), Taiwan, under grant MOST 109-2218-E-006-032.

**Conflicts of Interest:** The authors declare no conflict of interest.

## Abbreviations

The following abbreviations are used in this manuscript:

| | |
|---|---|
| GCP | generalized cyclic pursuit |
| MAS | multi-agent system(s) |
| SC | sliding mode control(ler) |
| ISC | integral sliding mode control(ler) |

## Appendix A. Circulant Matrix and Kronecker Product

A circulant matrix is a square matrix in which each row vector is rotated one element to the right relative to the preceding row vector. It has a specific eigenvalue repartition, and for (8)

$$\lambda_k = 1 - \exp\left(\mathbf{i}\frac{2\pi k}{n}\right), \quad k = 1, \cdots, n \tag{A1}$$

where $\lambda_k$ are the eigenvalues. The matrix (8) is a special case of Laplacian matrix, which represents, in formation control and graph theory, the communication graph among agents. The Kronecker product "$\otimes$" is an operation on two matrices of arbitrary size resulting in a block matrix. If $W$ is an $m \times n$ matrix whose elements are $w_{ij}$, $i = 1, ..., m$ and $j = 1, ..., n$, and $R$ is a $p \times q$ matrix, then the Kronecker product $W \otimes R$ is the $mp \times nq$ block matrix

$$W \otimes R = \begin{bmatrix} w_{11}R & w_{12}R & \dots & w_{1n}R \\ w_{21}R & w_{22}R & \dots & w_{2n}R \\ \vdots & \vdots & \ddots & \vdots \\ w_{m1}R & w_{m2}R & \dots & w_{mn}R \end{bmatrix} \tag{A2}$$

Suppose that $W$ and $R$ are square matrices of size $n$ and $p$ respectively. Let $\eta_1, \dots, \eta_n$ be the eigenvalues of $W$, and $\mu_1, \dots, \mu_p$ be the eigenvalues of $R$, then the eigenvalues of $W \otimes R$ are

$$\lambda_{ij} = \eta_i \mu_j, \quad i = 1, \dots, n. \ j = 1, \dots, p. \tag{A3}$$

See [35–38] for further explanations.

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

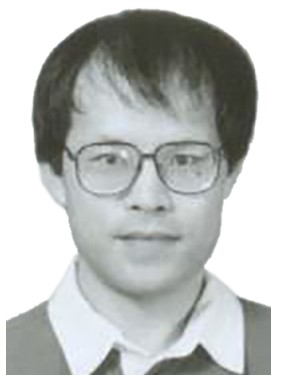

**Jyh-Ching Juang** received the B. S. and M. S. degrees from National Chiao-Tung University, Hsin-Chu, Taiwan, in 1980 and 1982, respectively, and the Ph.D. degree in electrical engineering from University of Southern California, Los Angeles, in 1987 and he is currently a fellow member of IEEE. He was with Lockheed Aeronautical System Company, Burbank before he joined the faculty of the Department of Electrical Engineering, National Cheng Kung University, Tainan, Taiwan in 1993. His research interests include satellite navigation and control, sensor networks, GNSS signal processing, and software-based receivers. He is coordinating a nano/micro-satellite development team at National Cheng Kung University.