# Peer review of "Integral Sliding Control Approach for Generalized Cyclic Pursuit Formation Maintenance"

_electronics, doi:10.3390/electronics10101217_

Round 1

Reviewer 1 Report

Generally I found both the article and the topic interesting.

However the whole article lacks some clarity both in the overall structure but especially also in the general definitions.

Particularly some basic definitions and especailly some paragraphs on the eigenvalues are not clear enough described, whcih is further complicated by additional Issues in English Grammar and not detailed enough description in particular where the eigenvalues and their calculations are introduced. 

Furtherore I think there is a limitation in clarity what is really novel in this context. There seem to be a number of minor differences in used terminology in the paper itself and also in the cited articles. Thus it is not always clear what the author try to convey and what is really novel.

For example gain is not clearly defined in the introduction, but then in the theorem definition it is stated that "gains" and "angle" are used interchangably.

Beyond this I found a number of minor issues:

                Line 23-24. The grammar in this sentence seems messed up. Did you mean: “. On another side, researchers have focused on non-circular pattern generation”?

Line 28-29: Its not really clear what is meant here with the sentence. Specificity as a word seems to be slightly off use here. Probably the author meant “specific properties” of circular formation ? Already the term “circular formation” is only loosely defined in line 21. But now also gains is not clearly defined. Gains in what. Altogether the sentence need to be clarified and cleared up. Maybe the example from the next sentence need to be put forward ?

Line 59-60. Grammar and time of the sentence is off. “can be reduced” ?

Line 61: “depend” instead of “depends”

Line 61-62: Unclear if gains is a values for all agents or for each of those agents.

Line 65-67: This is a very “uncertain” description. Could this be clarified. What is meant by “avoidance of uncertainties”. Additionally, ISC is defined as abbreviation in the abstract, but not in the text itself. There seems to be also some ambiguity here with the terms used and the terms used in the cited papers. Perhaps this might be improved?

Line 72: I would suggest to introduce the figure early on. Its easier to understand then.

Line 72 / equation 5:  I can follow until equation 5 is introduced. I fear this certainly needs a clarification on a) where this equation originates from and b) in regard to giving a more general definition of the eigenvalues here in their specific meaning in regard to the specific problem here.

This is also true in regard to figure 2, which needs a slightly more extensive description because it is not completely clear what is shown in the figure.

Line 87: Grammar

Line 93: “eigenvalues of (4) will possess two pairs of imaginary eigenvalues”. This does not seem to make sense to me. Really eigenvalues of eigenvalues ?

Author Response

Dear reviewer,

thank you for your answers and comments, we do appreciate them. We brought some corrections to our paper and we found it more convenient to summarize all our answers in the attached Word file.

Thanks in advance for your time and future comments

Best regards,

Antoine A. and Jyh-Ching J.

Reviewer 2 Report

please see attachment.

Author Response

(The authors gave the same response as above.)

Reviewer 3 Report

I appreciate the introduction into the subject to be very clear and concise. The mathematical explanations are useful too. I think it is important that you improved the results of the simulations by eliminating the chattering using a low pass filter and that the control effort is largely reduced in that manner. The target of more robust controllers is a target that could lead you to more interesting research. After reading your paper I think that your abstract could be improved. This is my opinions. The paper is an important one and I hope you will continue your research. Congratulations!

Author Response

(The authors gave the same response as above.)

Round 2

Reviewer 2 Report

Authors has revised the paper very well and reply the comments satisfactory.  It is accepted at my end.